# PeerJ

# Immune stimulation reduces sleep and memory ability in *Drosophila melanogaster*

Eamonn B. Mallon[1], Akram Alghamdi[2], Robert T.K. Holdbrook[1] and Ezio Rosato[3]

[1] Department of Biology, University of Leicester, Leicester, United Kingdom
[2] Department of Biology, Taif University, Saudi Arabia
[3] Department of Genetics, University of Leicester, Leicester, United Kingdom

## ABSTRACT

Psychoneuroimmunology studies the increasing number of connections between neurobiology, immunology and behaviour. We demonstrate the effects of the immune response on two fundamental behaviours: sleep and memory ability in *Drosophila melanogaster*. We used the Geneswitch system to upregulate peptidoglycan receptor protein (PGRP) expression, thereby stimulating the immune system in the absence of infection. Geneswitch was activated by feeding the steroid RU486, to the flies. We used an aversive classical conditioning paradigm to quantify memory and measures of activity to infer sleep. Immune stimulated flies exhibited reduced levels of sleep, which could not be explained by a generalised increase in waking activity. Immune stimulated flies also showed a reduction in memory abilities. These results lend support to *Drosophila* as a model for immune–neural interactions and provide a possible role for sleep in the interplay between the immune response and memory.

Corresponding author
Eamonn B. Mallon, ebm3@le.ac.uk

## INTRODUCTION

Psychoneuroimmunology, in vertebrates, studies the connections between neurobiology, immunology and behaviour (*Ader, Felten & Cohen, 1991*). These neural-immune interactions have also been found in invertebrates (*Demas, Adamo & French, 2011*). For example, immune response negatively affects learning and memory in bees (*Mallon, Brockmann & Schmid-Hempel, 2003*; *Gegear, Otterstatter & Thomson, 2006*; *Riddell & Mallon, 2006*; *Iqbal & Mueller, 2007*; *Alghamdi et al., 2008*). A tractable invertebrate model of these immune–neural links would provide a stimulus to this field (*Aubert, 2007*). The fruit fly, *Drosophila melanogaster*, has been tremendously helpful to the analysis of associative learning (*Kim, Lee & Han, 2007*) and immunity (*Lemaitre & Hoffmann, 2007*). In this paper we demonstrate immune-memory links in *Drosophila* and further expand the paradigm by showing immune-sleep interactions in flies.

Sleep is a resting state where the sleeper exhibits inattention to the environment and is usually immobile (*Siegel, 2003*). *Drosophila melanogaster* like vertebrates have been shown

to have a distinct sleep state. In flies, a sleep episode is defined as a period of immobility lasting five minutes or longer (*Hendricks et al., 2000*; *Shaw et al., 2000*). Such intervals are associated with reversible increases in arousal threshold, which can be further augmented following sleep deprivation (*Huber et al., 2004*), are associated with changes in brain electrical activity (*Nitz et al., 2002*; *van Alphen et al., 2013*), and are reduced by several drugs like caffeine and modafinil and are increased by antihistamines (*Hendricks et al., 2000*; *Shaw et al., 2000*). As in mammals, sleep deprivation leads to a rebound in quantity of sleep (*Shaw et al., 2000*).

Infections increase sleep in humans, most likely through induction of proinflammatory cytokines (*Bryant, Trinder & Curtis, 2004*). Fruit flies infected with gram-negative bacteria also show increased sleep (*Kuo et al., 2010*). Another study found flies infected with gram-positive bacteria slept less (*Shirasu-Hiza et al., 2007*).

Here, we activated the immune system non-pathogenically (*Moret & Schmid-Hempel, 2000*; *Mallon, Brockmann & Schmid-Hempel, 2003*; *Riddell & Mallon, 2006*; *Alghamdi et al., 2008*; *Richard, Aubert & Grozinger, 2008*). This separates the effect of the immune response from any direct effect of the pathogen, for example, parasite manipulation of the host (*Adamo & Webster, 2013*). We used Geneswitch (*Osterwalder et al., 2001*) to up-regulate peptidoglycan receptor protein LCa (PGRP-Lca) in adult flies. PGRP-Lca is a pattern recognition protein that recognizes DAP type peptidoglycan which is found on Gram negative and Gram positive bacteria setting off the IMD immune pathway and leading to the expression of antimicrobial peptides (*Gottar et al., 2002*). Geneswitch is activated in the presence of the steroid RU486. We used an aversive classical conditioning paradigm to measure memory abilities of flies (*Mery & Kawecki, 2005*). Sleep was measured using the *Drosophila* Activity Monitoring System 2 (DAMS2, Trikinetics, Waltham, MA).

## METHODS AND MATERIALS

The Geneswitch line $w^{1118}; P\{w^{+mW.hs} = Switch1\}bun^{Switch1.32}$ (hereafter referred to as $GS1.32$) drives expression of RU486-activated GAL4 in adult fat bodies (*Gottar et al., 2002*) (http://flystocks.bio.indiana.edu). The three genotypes used were $GS1.32 > PGRP-Lca(w^{1118}; GS1.32/+; UAS-PGRP-Lca/+)$, and the control genotypes $GS1.32/+ (w^{1118}; GS1.32/+; +/+)$ and $+/PGRP-Lca(+/+; UAS-PGRP-Lca/+)$.

Flies were maintained in vials containing agar, sugar, and Brewer's yeast media in a 12 h: 12 h light: dark cycle at 25 °C. This food was also used during all behavioural assays. Males and females were selected at eclosion and flies were 1–3 days old at the beginning of the experiment. Both sexes were used for the memory assay and sleep assay (*Isaac et al., 2010*).

### Geneswitch

In the Geneswitch system, the DNA binding domain of the GAL4 protein is fused to the activation moiety of p65 through a mutant progesterone receptor ligand binding domain. Thus, Geneswitch is a chimeric ligand-stimulated activator of transcription. In the absence of ligand, the Geneswitch is in the "off" state. In the presence of the antiprogestin RU486 the Geneswitch molecule changes to an active conformation, in which it binds, as a dimer, to UAS sequences and activates transcription of downstream genes. In flies, Geneswitch

mediated expression can be detectable 3–5 h after feeding on RU486, reaching maximal levels 21–48 h later (*Osterwalder et al., 2001*; *Roman et al., 2001*).

20 ml of RU486 (Sigma Aldrich) 10 mM stock solution (0.13 g of RU486 in 32 ml of 80% ethanol) was mixed with 980 ml molten *Drosophila* food (200 μM final concentration). For the memory assay, flies were fed for two days with RU486 before the start of the training and returned to the RU486 food after training. For the sleep assay, flies were placed in vials containing RU486 food for two days to allow feeding. After two days flies were immediately loaded into tubes containing more of the RU486 food. For all lines we have flies fed with RU486 and genetically identical animals cultured on fly medium supplemented with an equal amount of vehicle (80% ethanol) that lacked RU486.

## Memory assay

Each sample was a single sex group of 50 adult flies. This memory assay was described previously (*Mery & Kawecki, 2005*). Conditioning consisted of 5 training sessions separated by 20 min intervals. In each training session flies were first exposed for 30 s to one odorant simultaneously with mechanical shock delivered every 5 s. This period was followed by a 60 s rest period (no odour and no shock). Then, for 30 s another odorant was delivered, without shock. Flies were either conditioned against 3-octanol or 4-methylcyclohexanol (both 0.6 ml/l of paraffin).

24 h after the conditioning period flies were transported to the choice point of a T-maze, where they were allowed to choose between the two odors for 60 s. The memory score was the proportion of individuals choosing the correct odour, i.e., not the one they were trained against. One hundred and fifteen replicates were carried out, distributed between the genotype, sex, RU486 (presence/absence) and odour used.

## Sleep assay

Fly locomotor activity was monitored by the *Drosophila* Activity Monitoring System 2 (DAMS2, Trikinetics, Waltham, MA), at 25 °C, continuously for seventy-two hours under a 12:12 light:dark cycle. Output from DAMS2 was the number of times a fly crossed an infrared beam in a given 1 min period (bin). A sleep episode (bout) was defined as 5 or more consecutive bins of immobility. 345 flies were tested, divided between genotype, sex and RU486 (presence/absence) (mean = 28.75 flies per group).

## Data analysis for sleep assay

The DAMS2 output was converted to five measures; (1) Sleepbins per hour: number of minutes when a fly is asleep in an hour, (2) Mean waking activity: the mean number of times a fly crosses the beam in 1 min bins that are classified as 'waking', (3) Bouts of sleep: the number of sleep episodes per hour, (4) Average sleep bout duration, and (5) Sleep latency: the amount of time since lights off before the first sleep bout occurs.

A four way ANOVA was performed. The independent variables were genotype, RU486 (presence/absence), sex and time. The important term here is an interaction term between genotype and RU486. If this was significant, the genotypes responded differently to the treatments. To discover which genotypes were significantly different two further

four-way ANOVAs were performed, one for genotypes GS1.32 > PGRP-Lca vs. GS1.32/+ and one for genotypes GS1.32 > PGRP-Lca vs. +/PGRP-Lca. If the interaction terms in both these ANOVAs are significant GS1.32 > PGRP-Lca (the immune stimulated genotype) responses differently to the control genotypes. Using a Bonferroni correction the significance level $\alpha$ was reduced to 0.0033 (0.05/15) for the sleep data, as there were three separate ANOVAs carried out for each of 5 measures. All analysis was carried out using R 3.01 (*R Core Team, 2013*).

### Zone of inhibition assay

Our treatment line had previously been shown to upregulate the immune response (*Gottar et al., 2002*). However we used the zone of inhition assay to confirm increased immune response in our treated flies. This assay measures antibacterial activity: it is based on the ability of immune proteins to inhibit bacterial growth when placed onto an agar plate seeded with bacteria (*Arthrobacteur globiformis* 125 μl of an overnight culture per 50 ml of agar). Thirty seven GS1.32 > PGRP-Lca flies, 17 fed RU486 and 20 not fed RU486 were used. Each fly was homogenized in 30 μl of ringer solution. Five microlitres of the supernatant from the centrifuged solution (1300 g for 10 min at 4 °C) were pipetted into a hole on the agar plate. This was incubated for 48 hrs (30 °C). The resultant ZOI were measured as the mean of three diameters.

### RESULTS

Raw data is available on figshare (http://dx.doi.org/10.6084/m9.figshare.1030499). Feeding RU486 to GS1.32 > PGRP-Lca flies increased their antibacterial activity by 26% (mean +/− standard error: RU486+ = 5.85 mm +/−0.25, RU486−= 4.65 +/−0.45, $t = -2.3263$, $df = 29.202$, $p = 0.02715$).

### Immune stimulation effects on memory

Genotype had a significant effect on memory score ($F_{2,109} = 22.46$, $p < 0.0001$). The main effects for sex, whether RU486 was used, and odour used were not significant. There was however a significant interaction between genotype and whether RU486 was used ($F_{2,109} = 5.76$, $p = 0.0042$). GS1.32 > PGRP-Lca flies, showed a 11.4% decrease in memory scores when fed RU468 relative to those not fed RU468 of the same genotype (Tukey HSD: GS1.32 > PGRP-Lca + RU468 vs. GS1.32 > PGRP-Lca − RU468 $p = 0.0418$, GS1.32/++ RU486 vs. GS1.32/+ − RU486 $p = 0.9578$, +/PGRP-Lca + RU486 vs. +/PGRP-Lca − RU486 $p = 0.6784$). See Fig. 1. As feeding RU486 to GS1.32 > PGRP-Lca flies leads to an increased immune response, immune stimulation decreases memory scores.

### Immune stimulation effects on sleep

The complete ANOVAs for all measures are available in the Supplemental Information. Immune stimulated flies (GS1.32 > PGRP-Lca fed with RU486) showed a decrease in sleep relative to controls (Complete 4-way ANOVA genotype:ru $F_{2,23976} = 158.74$, $p < 0.00001$, GS1.32 > PGRP-Lca vs. GS1.32/+: $F_{1,16632} = 282.37$, $p < 0.00001$, GS1.32 > PGRP-Lca vs. +/ PGRP-Lca: $F_{1,15840} = 11.82$, $p = 0.00059$). See Figs. 2 and 3. Table 1 gives the means and

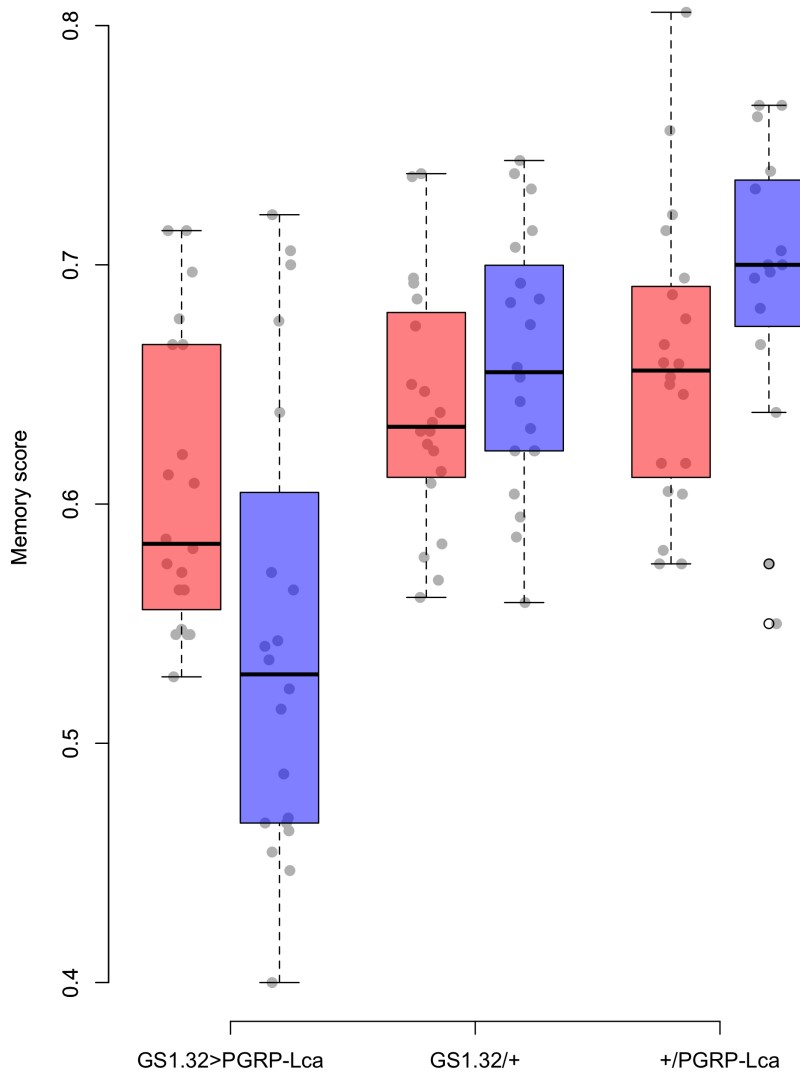

**Figure 1 Memory score for each geneotype.** Memory score is the proportion of flies that choose the odour they were not trained against. The blue boxes represent the mean memory score for the RU486− flies. The red boxes represent the RU486+ flies. The grey dots are the individual data points.

standard errors. As there was no significant interaction between sex, genotype and whether RU486 was used (Complete 4-way ANOVA genotype:ru $F_{2,23976} = 20.79$, $p < 0.00001$, GS1.32 > PGRP-Lca vs. GS1.32/+: $F_{1,16632} = 32.75$, $p < 0.00001$, GS1.32 > PGRP-Lca vs. +/PGRP-Lca: $F_{1,15840} = 0.01$, $p = 0.90374$) a single sex analysis is not shown. However dividing into males and females did not alter the significant effect of immune response on sleep.

Immune response had no effect on mean waking activity (Complete 4-way ANOVA genotype:ru $F_{2,23976} = 21.96$, $p < 0.00001$, GS1.32 > PGRP-Lca vs. GS1.32/+: $F_{1,16632} = 0.49$, $p = 0.4858$, GS1.32 > PGRP-Lca vs. +/PGRP-Lca: $F_{1,15840} = 39.18$, $p < 0.00001$). See Figs. 4 and 5. Table 2 gives the means and standard errors for mean waking activity.

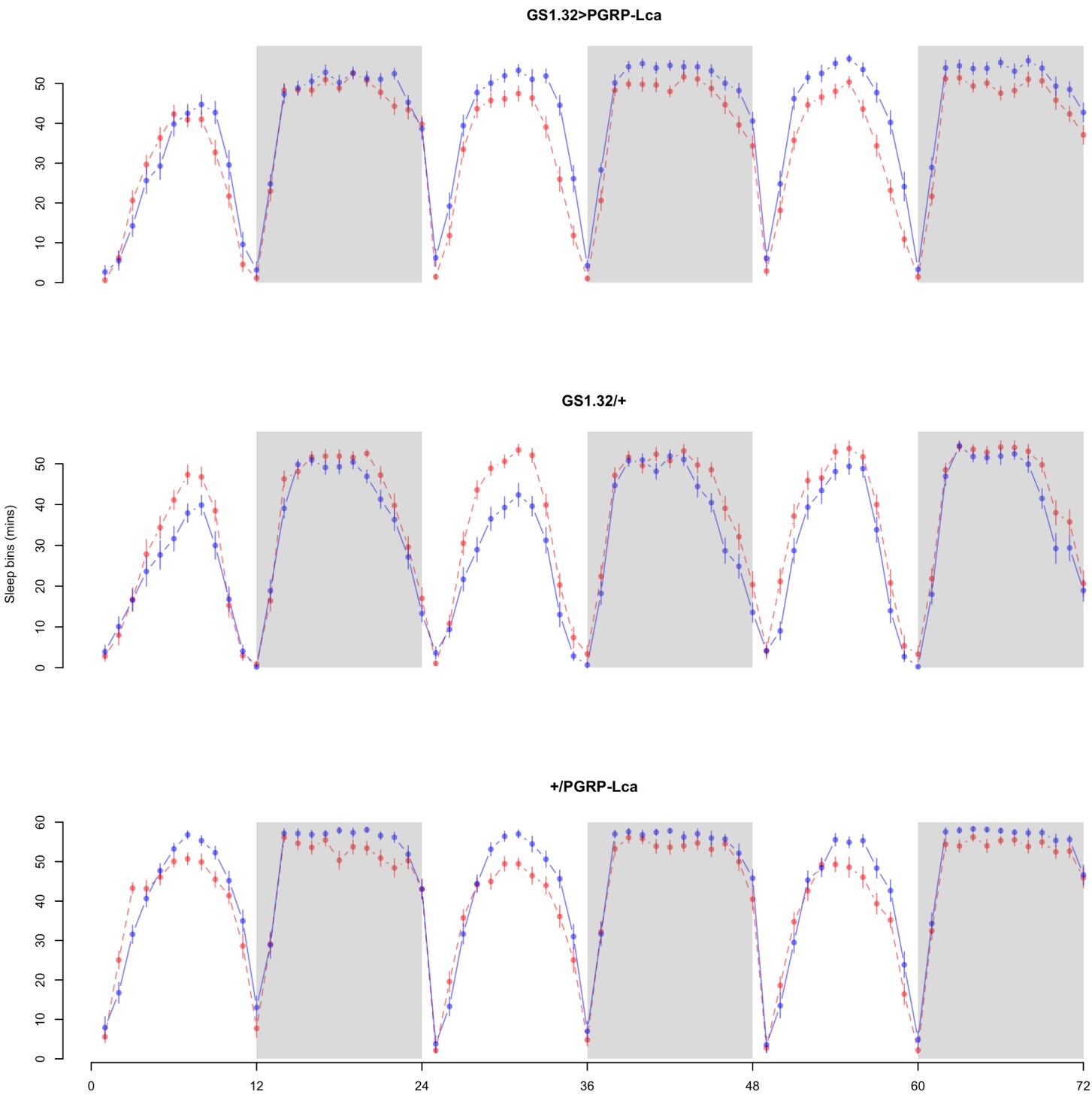

**Figure 2** **Male sleepbins.** Sleepbins for the males for each genotype. The blue points represent the means for the RU486− flies. The red points represent the means of RU486+ flies. Error bars are standard errors. The shaded times are night (lights off).

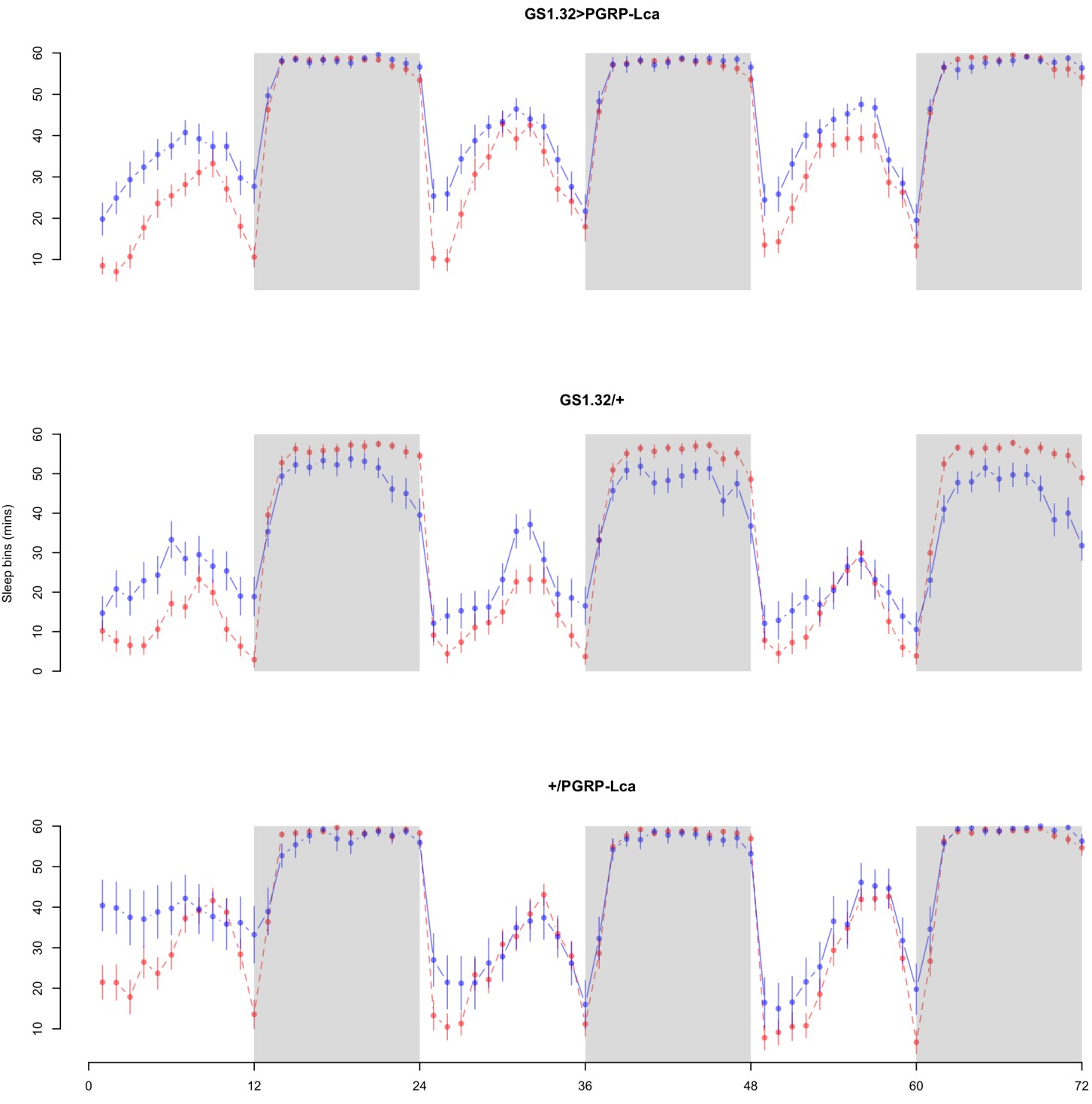

**Figure 3 Female sleepbins.** Sleepbins for the females for each genotype. The blue points represent the means for the RU486− flies. The red points represent the means of RU486+ flies. Error bars are standard errors. The shaded times are night (lights off).

**Table 1  Sleep bins.** Means and standard errors of the number of sleep bins per hour for the 12 groups of flies used in the sleep experiment.

| Genotype | RU486− | RU486+ | % change |
|---|---|---|---|
| **Male** | | | |
| GS1.32 > PGRP-Lca | 41.15 +/− 0.43 | 36.41 +/− 0.41 | 12% decrease |
| GS1.32/+ | 31.51 +/− 0.45 | 35.89 +/− 0.45 | 14% increase |
| +/PGRP-Lca | 45.31 +/− 0.41 | 42.45 +/− 0.40 | 6% decrease |
| **Female** | | | |
| GS1.32 > PGRP-Lca | 45.84 +/− 0.41 | 41.05 +/− 0.46 | 10% decrease |
| GS1.32/+ | 33.45 +/− 0.56 | 33.11 +/− 0.53 | 1% decrease |
| +/ PGRP-Lca | 43.67 +/− 0.66 | 40.67 +/− 0.50 | 7% decrease |

**Table 2  Mean waking activity.** Means and standard errors of the number of times a fly crosses the beam per minute during 'waking' period (Mean waking activity) for the 12 groups of flies used in the sleep experiment.

| Genotype | RU486− | RU486+ | % change |
|---|---|---|---|
| **Male** | | | |
| GS1.32 > PGRP-Lca | 1.888 +/− 0.033 | 2.324 +/− 0.027 | 23% increase |
| GS1.32/+ | 2.252 +/− 0.024 | 2.315 +/− 0.032 | 3% increase |
| +/ PGRP-Lca | 1.725 +/− 0.039 | 2.379 +/− 0.046 | 38% increase |
| **Female** | | | |
| GS1.32 > PGRP-Lca | 1.790 +/− 0.046 | 1.452 +/− 0.028 | 19% decrease |
| GS1.32/+ | 1.817 +/− 0.042 | 1.905 +/− 0.041 | 5% increase |
| +/ PGRP-Lca | 1.405 +/− 0.060 | 1.398 +/− 0.031 | 1% decrease |

Immune response had a significant effect on number of sleep bouts (Complete 4-way ANOVA genotype:ru $F_{2,23976} = 69.9$, $p < 0.00001$, GS1.32 > PGRP-Lca vs. GS1.32/+: $F_{1,16632} = 013.08$, $p = 0.0003$, GS1.32 > PGRP-Lca vs. +/PGRP-Lca: $F_{1,15840} = 71.52$, $p < 0.00001$). See Figs. 6 and 7. Table 3 gives the means and standard errors for number of sleep bouts. This result is difficult to interpret as RU486+ GS1.32 > PGRP-Lca flies have more disturbed sleep than RU486+ GS1.32/+ flies but less disturbed sleep than RU486+ +/PGRP-Lca flies.

Immune response had no effect on sleep bout duration (Complete 4-way ANOVA genotype:ru $F_{2,23976} = 203.49$, $p < 0.00001$, GS1.32 > PGRP-Lca vs. GS1.32/+: $F_{1,16632} = 290.48$, $p < 0.00001$, GS1.32 > PGRP-Lca vs. +/PGRP-Lca: $F_{1,15840} = 7.78$, $p = 0.0053$ (Not significant)). See Figs. 8 and 9. Table 4 gives the means and standard errors for sleep bout duration.

Immune response had no effect on sleep latency (Complete 4-way ANOVA genotype:ru $F_{2,925} = 1.78$, $p = 0.16935$). See Figs. 10 and 11. Table 5 gives the means and standard errors for sleep latency.

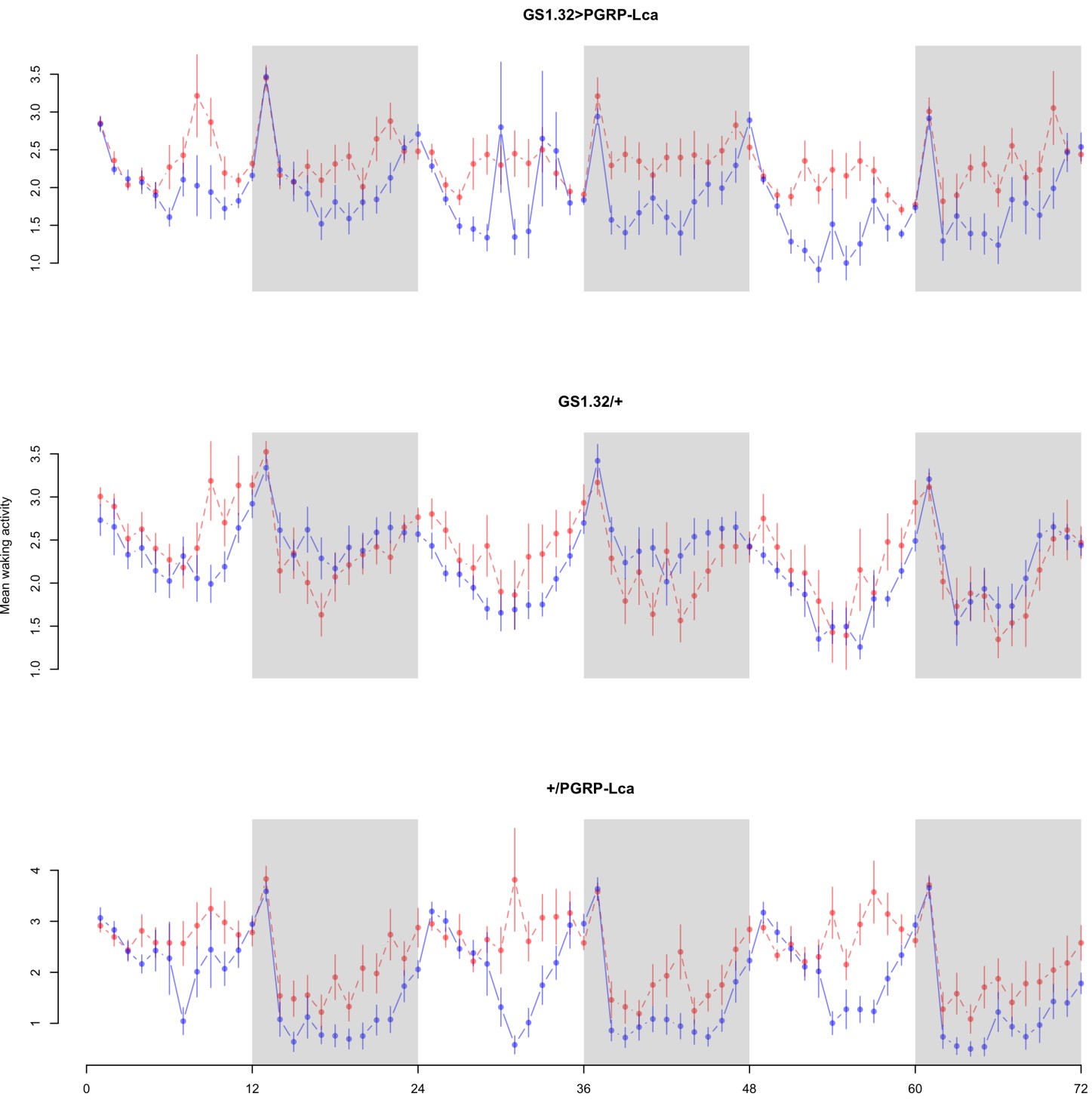

**Figure 4  Male mean waking activity.** Mean waking activity for the males for each genotype. The blue points represent the means for the RU486−
flies. The red points represent the means of RU486+ flies. Error bars are standard errors. The shaded times are night (lights off).

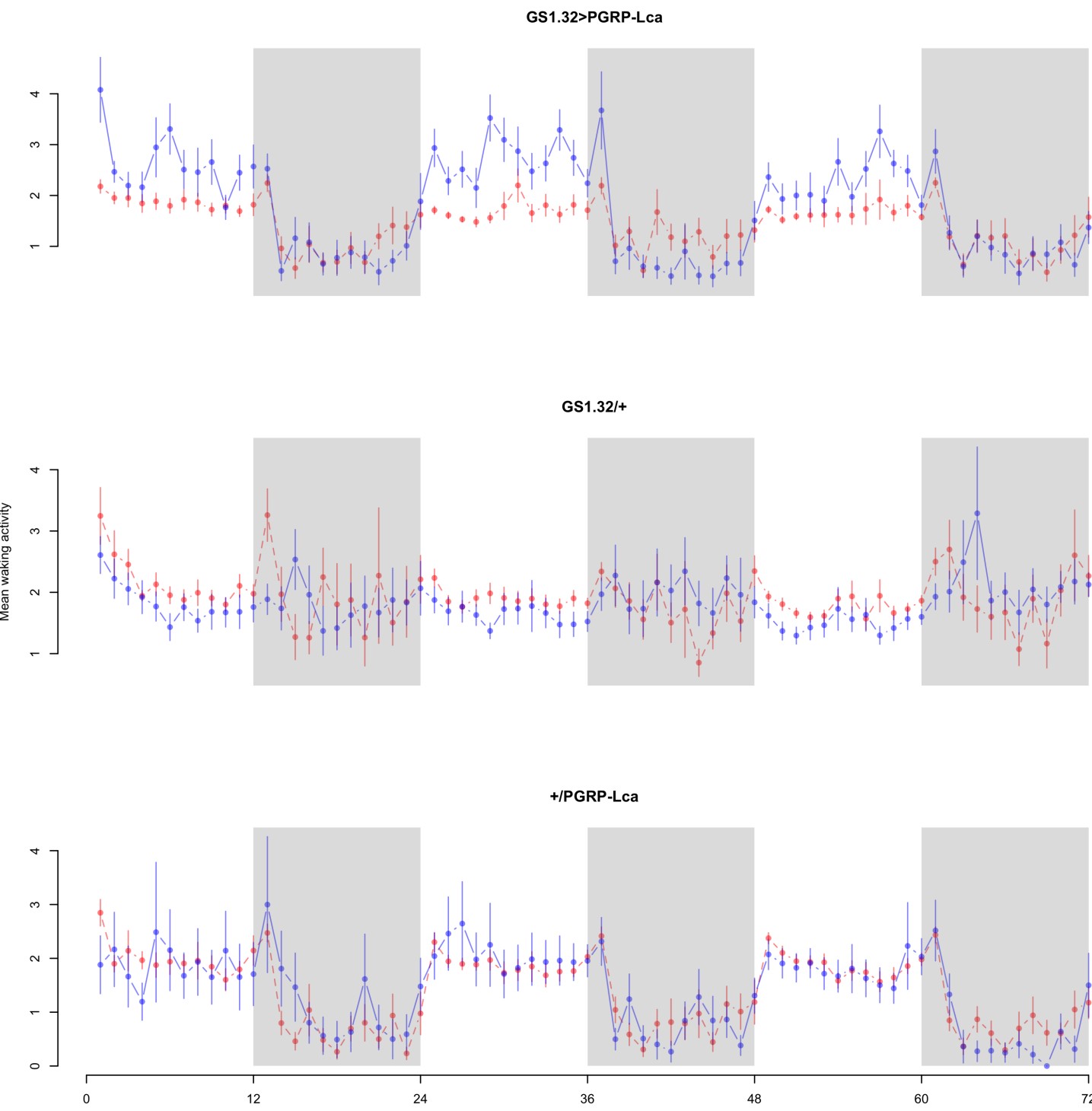

**Figure 5 Female mean waking activity.** Mean waking activity for the females for each genotype. The blue points represent the means for the RU486− flies. The red points represent the means of RU486+ flies. Error bars are standard errors. The shaded times are night (lights off).

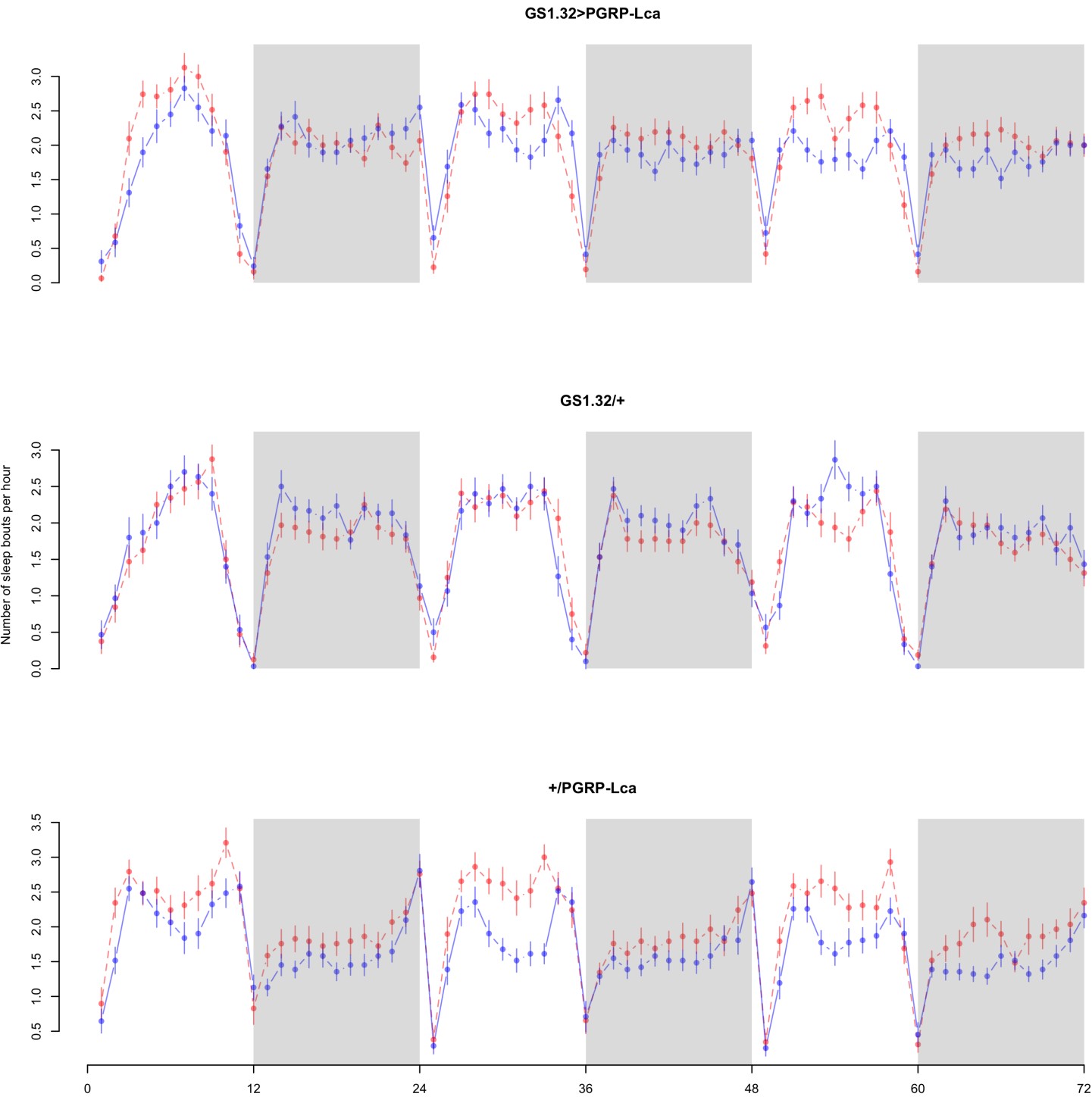

**Figure 6 Male number of sleep bouts.** Number of sleep bouts per hour for the males for each genotype. The blue points represent the means for the RU486− flies. The red points represent the means of RU486+ flies. Error bars are standard errors. The shaded times are night (lights off).

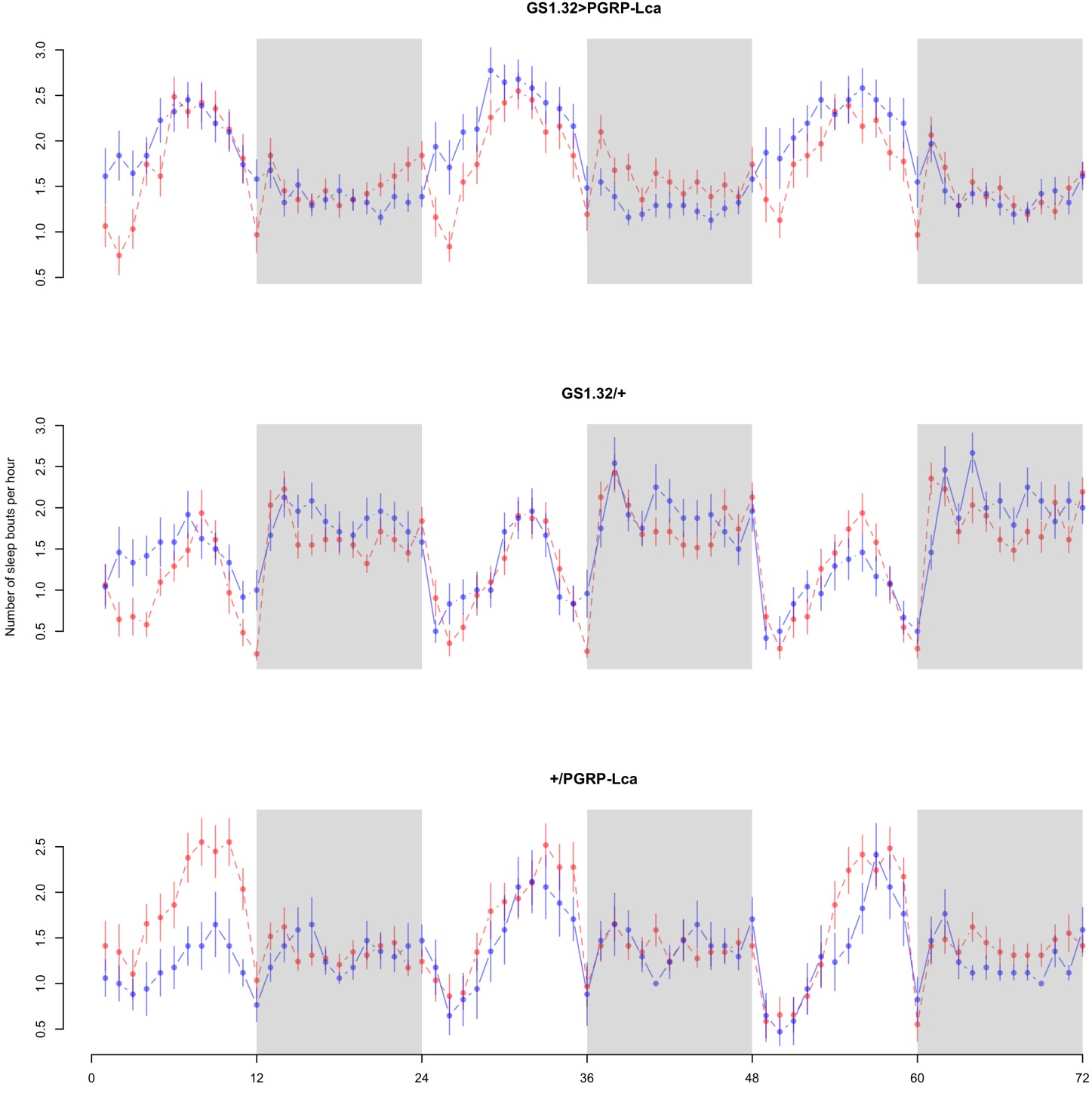

**Figure 7 Female number of sleep bouts.** Number of sleep bouts per hour for the females for each genotype. The blue points represent the means for the RU486− flies. The red points represent the means of RU486+ flies. Error bars are standard errors. The shaded times are night (lights off).

**Table 3 Sleep bouts.** Means and standard errors of sleep bouts per hour for the 12 groups of flies used in the sleep experiment.

| Genotype | RU486− | RU486+ | % change |
|---|---|---|---|
| **Male** | | | |
| GS1.32 > PGRP-Lca | 1.849 +/− 0.024 | 1.955 +/− 0.025 | 6% increase |
| GS1.32/+ | 1.779 +/− 0.026 | 1.695 +/− 0.025 | 5% decrease |
| +/PGRP-Lca | 1.673 +/− 0.023 | 2.021 +/− 0.026 | 21% increase |
| **Female** | | | |
| GS1.32 > PGRP-Lca | 1.755 +/− 0.024 | 1.661 +/− 0.023 | 5% decrease |
| GS1.32/+ | 1.555 +/− 0.029 | 1.419 +/− 0.025 | 9% decrease |
| +/PGRP-Lca | 1.324 +/− 0.030 | 1.528 +/− 0.025 | 15% increase |

**Table 4 Sleep bout duration.** Means and standard errors of sleep bout duration for the 12 groups of flies used in the sleep experiment.

| Genotype | RU486− | RU486+ | % change |
|---|---|---|---|
| **Male** | | | |
| GS1.32 > PGRP-Lca | 25.96 +/− 0.44 | 20.05 +/− 0.34 | 23% decrease |
| GS1.32/+ | 17.85 +/− 0.36 | 23.27 +/− 0.42 | 30% increase |
| +/ PGRP-Lca | 33.69 +/− 0.48 | 26.74 +/− 0.45 | 21% decrease |
| **Female** | | | |
| GS1.32 > PGRP-Lca | 34.17 +/− 0.49 | 29.24 +/− 0.48 | 14% decrease |
| GS1.32/+ | 23.21 +/− 0.54 | 23.35 +/− 0.49 | 1% increase |
| +/ PGRP-Lca | 37.22 +/− 0.73 | 31.29 +/− 0.54 | 16% decrease |

**Table 5 Sleep latency.** Means and standard errors of sleep latency for the 12 groups of flies used in the sleep experiment.

| Genotype | RU486− | RU486+ | % change |
|---|---|---|---|
| **Male** | | | |
| GS1.32 > PGRP-Lca | 27.56 +/− 1.19 | 34.68 +/− 1.68 | 26% increase |
| GS1.32/+ | 33.77 +/− 2.11 | 36.33 +/− 1.68 | 8% increase |
| +/ PGRP-Lca | 29.81 +/− 1.81 | 27.93 +/− 1.43 | 6% decrease |
| **Female** | | | |
| GS1.32 > PGRP-Lca | 18.22 +/− 3.67 | 10.79 +/− 0.83 | 41% decrease |
| GS1.32/+ | 34.94 +/− 4.58 | 20.34 +/− 1.18 | 42% decrease |
| +/ PGRP-Lca | 57.11 +/− 16.54 | 31.04 +/− 3.48 | 46% decrease |

## DISCUSSION

Immune stimulated adult flies exhibit reduced levels of sleep both during day and night. Immune stimulation also leads to a reduction in memory abilities.

The reduction in sleep cannot be explained simply in terms of a generalised increase in activity. Stimulating the immune response had no effect on mean waking activity during the day or night, but immune-stimulated flies slept less than the non-stimulated controls.

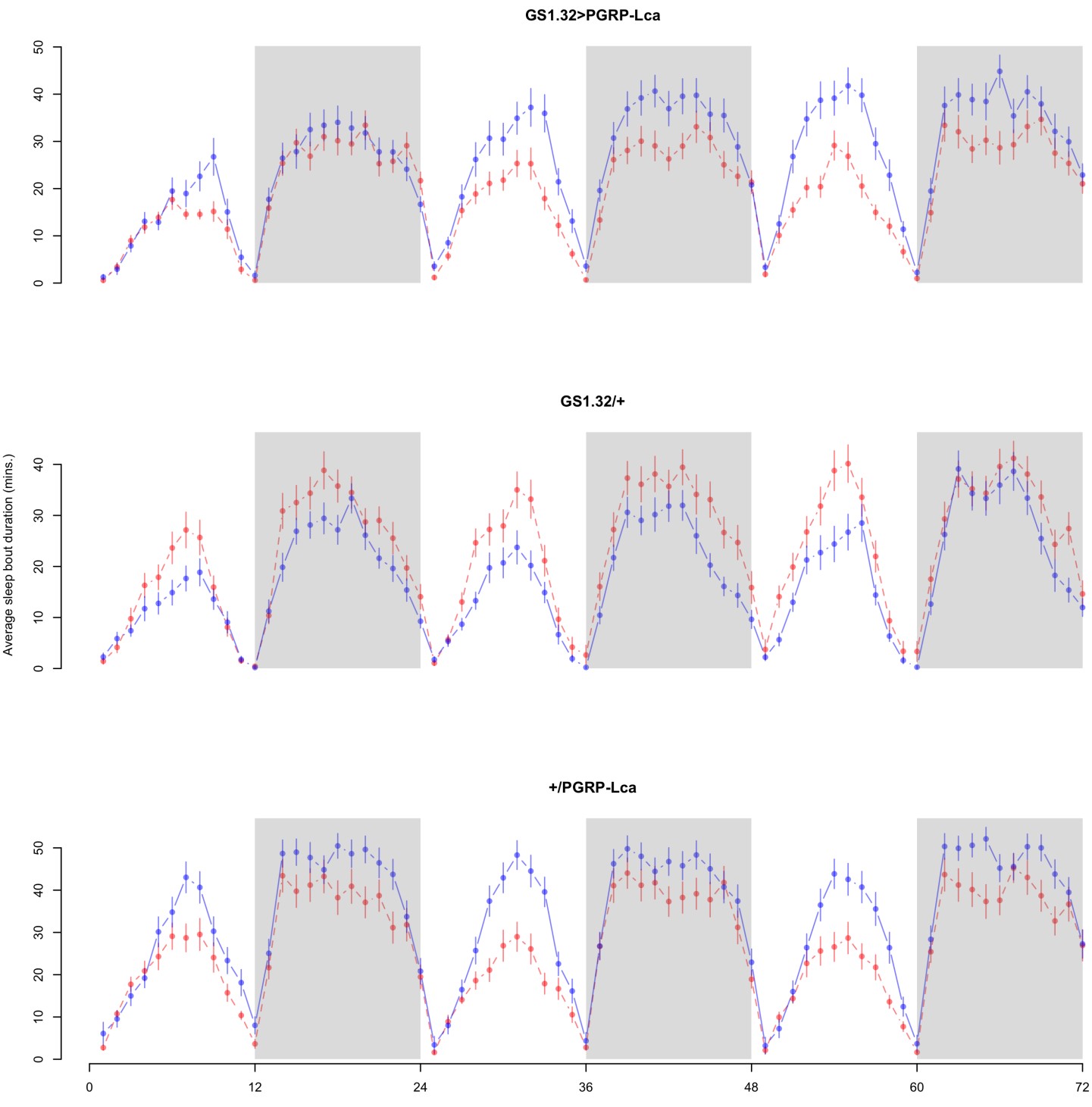

**Figure 8 Male sleep bout duration.** Sleep bout duration for the males for each genotype. The blue points represent the means for the RU486— flies. The red points represent the means of RU486+ flies. Error bars are standard errors. The shaded times are night (lights off).

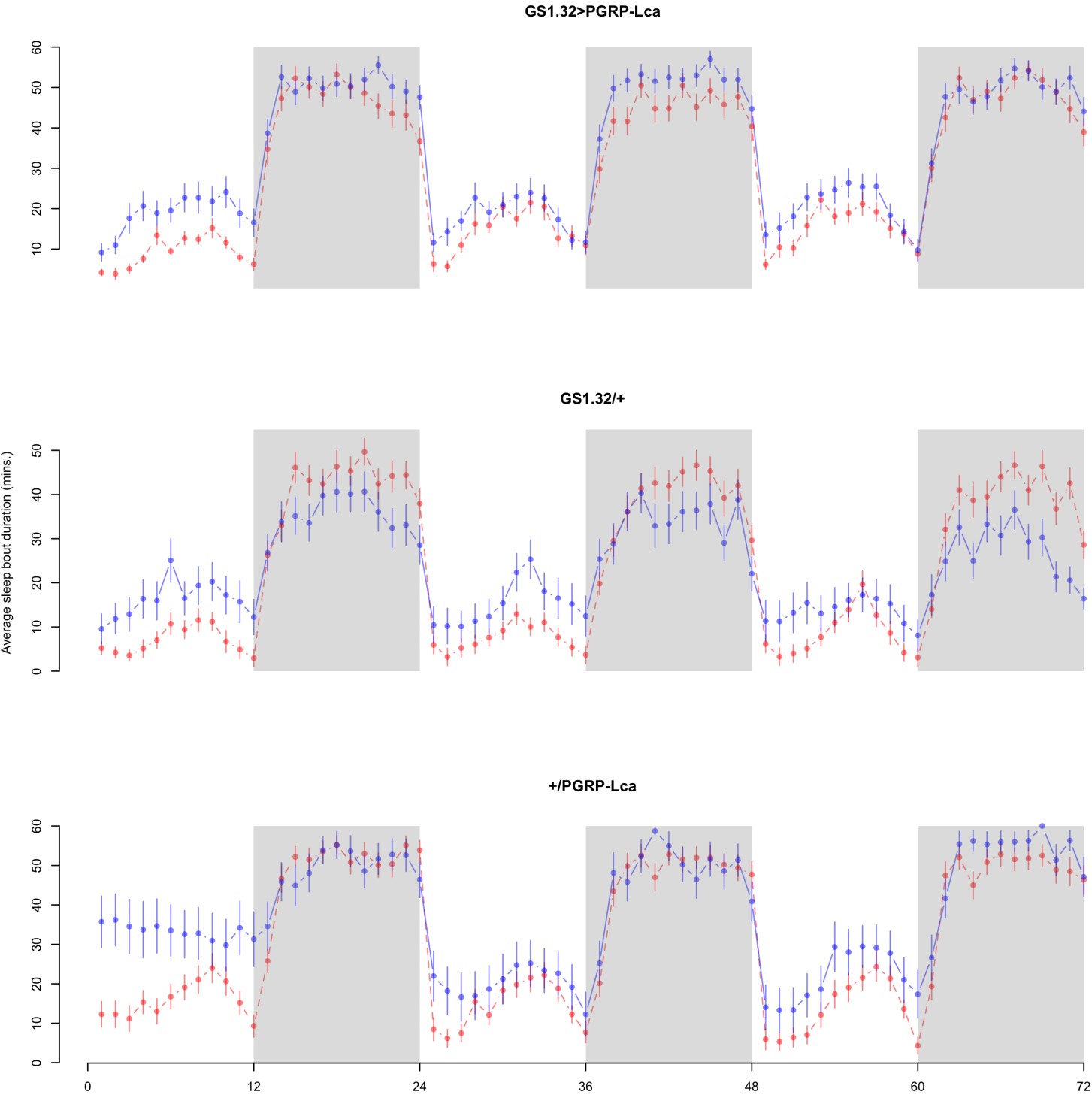

**Figure 9 Female sleep bout duration.** Sleep bout duration for the females for each genotype. The blue points represent the means for the RU486−
flies. The red points represent the means of RU486+ flies. Error bars are standard errors. The shaded times are night (lights off).

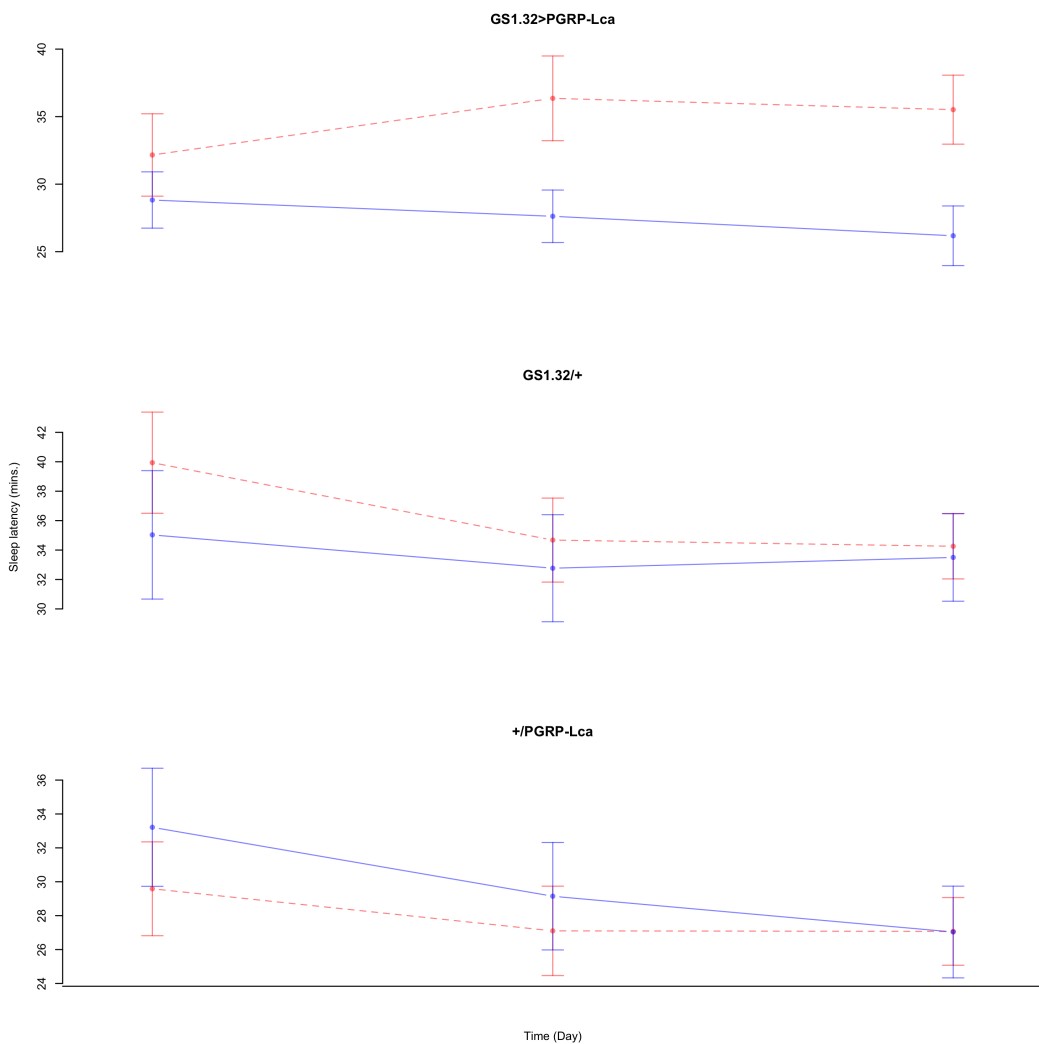

**Figure 10 Male sleep latency.** Sleep latency for the males for each genotype. The blue points represent the means for the RU486− flies. The red points represent the means of RU486+ flies. Error bars are standard errors.

As we did not test for the effects of immune response on olfactory sensitivity itself, we cannot say categorically that our effect on memory was not due to a decrease in the flies' ability to differentiate odours. However, experiments carried out on honeybees (*Mallon, Brockmann & Schmid-Hempel, 2003*) and bumblebees (*Riddell & Mallon, 2006*) found that the immune response did not effect their olfaction.

Our sleep results are difficult to relate to previous studies that found an effect of infection on sleep in Drosophila. Shirasu-Hiza showed a decrease in sleep after gram-positive bacterial infections (*Shirasu-Hiza et al., 2007*). However *Kuo et al. (2010)* found that when they infected flies with gram-negative bacteria, the flies slept more. These two results are difficult to compare as the experiments differed in numerous methodical aspects, e.g., strength of infection, lighting paradigm, when the phenotype was measured, etc.

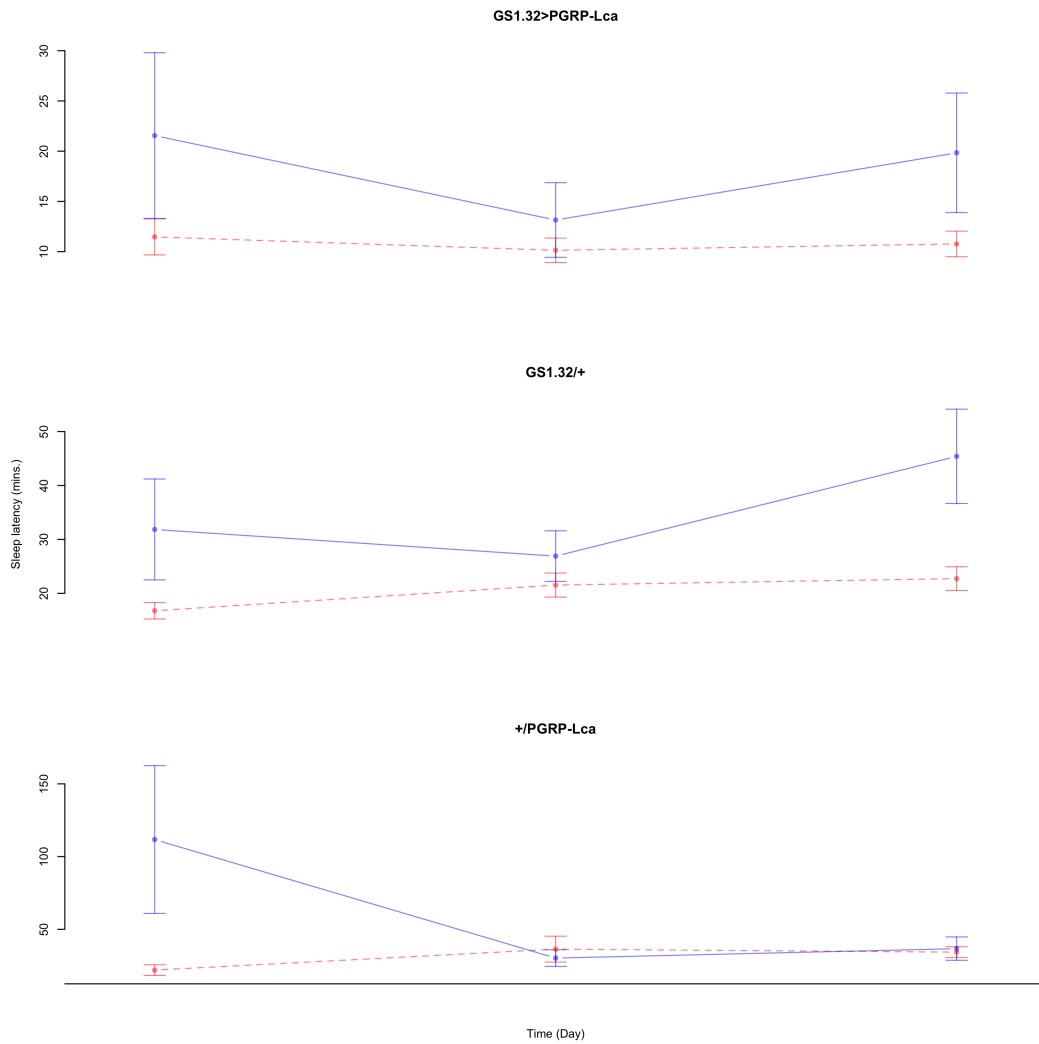

**Figure 11 Female sleep latency.** Sleep latency for the females for each genotype. The blue points represent the means for the RU486− flies. The red points represent the means of RU486+ flies. Error bars are standard errors.

Although Imd is one of the canonical immune pathways in insects, over-expression of the Imd pathway can also lead to apoptosis (*Georgel et al., 2001*; *Leulier et al., 2003*). It cannot be excluded that our results could be caused by a side effect: apoptosis of the fat body by the Imd pathway rather than its main effect of immune response. This will be examined in future work.

We have shown that immune response decreases sleep and memory in *Drosophila melanogaster.* We propose a possible link between all three systems as an interesting area for future research. One of the main hypotheses on sleep function is that sleep periods are favourable for brain plasticity and in the adult brain for learning and memory (*Maquet, 2001*). Like humans, flies with a fragmented sleep show impaired learning compared with flies with consolidated sleep (*Seugnet et al., 2008*). Flies also exhibit decreases in learning after 6 or 12 h of sleep deprivation (*Seugnet et al., 2008*). We propose sleep

as an intermediate between immunity and memory. We hypothesise that it is not the activation of the immune system *per se* that affects memory in flies, but rather that immune stimulation reduces the length and quality of sleep that in turn, reduces memory ability. However, with our current data, we cannot exclude that in flies the level of immune activation has a direct effect on memory.

Our results adds to the evidence for *Drosophila* as a model for immune–neural interactions. As well as the potential use as a model for mammalian neural-immune links, this work has direct impact on recent concern for insect foragers and the role of multiple stressors in their decline (*Gill, Ramos-Rodriguez & Raine, 2012*).

## ACKNOWLEDGEMENTS

Thanks to Dr. Frederic Mery (CNRS, Gif Sur Yvette) for initial discussions about setting up the memory assay. Thanks to E Green (Genetics, University of Leicester) for use of the excel plugin, Befly, to calculate sleep measures.

### Funding

ER and AA were funded by BBSRC grant BB/H018093/1 and a Saudi government scholarship respectively. The funders had no role in study design, data collection and analysis, decision to publish, or preparation of the manuscript.

### Grant Disclosures

The following grant information was disclosed by the authors:
BBSRC: BB/H018093/1.

### Competing Interests

The authors declare there are no competing interests.

### Author Contributions

- Eamonn B. Mallon conceived and designed the experiments, analyzed the data, wrote the paper, prepared figures and/or tables, reviewed drafts of the paper.
- Akram Alghamdi conceived and designed the experiments, performed the experiments, analyzed the data, wrote the paper, prepared figures and/or tables, reviewed drafts of the paper.
- Robert T.K. Holdbrook performed the experiments, reviewed drafts of the paper.
- Ezio Rosato conceived and designed the experiments, performed the experiments, contributed reagents/materials/analysis tools, wrote the paper, reviewed drafts of the paper.

### Data Deposition

The following information was supplied regarding the deposition of related data:
Figshare: http://dx.doi.org/10.6084/m9.figshare.1030499.

## Supplemental Information

Supplemental information for this article can be found online at http://dx.doi.org/10.7717/peerj.434.

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
