# Peer review of "Immune stimulation reduces sleep and memory ability in Drosophila melanogaster"

_PeerJ, doi:10.7717/peerj.434_

## Round 0.1 · original submission · Major Revisions

As you will see, two of the reviews indicated accept with minor revisions and a third indicated rejection. In reviewing the specific comments and criticisms, I think your work is suitable for publication, subject to addressing concerns raised in the reviews. A common theme in all the reviews is the need for more details and justification on methods. Also, I found the suggestions of the first reviewer compelling and useful, particular in the notion that you should, in so far as you can, let your data speak for itself. For instance, rather than saying that this work "establishes" D. melanogaster as a model, just say that your work "demonstrates" or "shows that D. melanogaster is suitable as". This suggestion may strike you as trivial, but I think your word choices and tone in how you present the implications of your findings underlies the comments and reactions of the reviewers to your paper.

As an author and editor, I frequently have difficulty distinguishing what is meant by "minor" vs. "major" revisions. In this instance, I indicated "major" revision because there are a large number of points raised in the reviews that need to be addressed, and additional presentation of data/details on non-significant finds may be warranted. As you make your revisions, please pay particular attention to addressing questions on dose which were identified by two of the reviewers.

Reviewer 1 ·

Basic reporting

The authors provide sufficient background to justify an interesting study. However, there were many shortcomings. Some were conceptual, and more importantly, a lot of the data on which the authors’ conclusions were based were not shown or even reported in the text. The statistical approach was misleading (see below), the figures and methods were not adequately described. These issues are listed below:

1. A “zone of inhibition” assay is described as a measure of immune function in flies over-expressing PGRP-Lca. The authors report a “26 %” increase in antibacterial activity, but do not show the data. It is assumed that a comparison was made between the RU486 and vehicle control condition in GS1.32>PGRP-Lca flies. However, no control assays were performed in the parent strains, leaving open the possibility that RU486 may have had non-specific effects on bacterial growth in this assay. More detailed methods are needed, especially with respect to analysis, control experiments on parent lines should be performed, and the data should be shown.

2. Figure 2 is not well described. Does each data point represent a mean across a group? If so, how many flies were in each group? Why are there no error bars?

3. Some effect of RU486 is reported on number of sleep bouts, which the authors conclude supports the idea that sleep is fragmented. However, this effect was very weak, was restricted to the night-time, and the data are not shown or even reported. The authors should have also evaluated average sleep bout duration.

4. The statement in lines 66-68 (introduction) is conceptually incorrect. The authors suggest that the finding that infection reduces sleep (Shirasu-Hiza et al, 2007) is in agreement with microarray analyses which showed that immune-related genes increase with sleep deprivation. Generally, molecular components that increase with prolonged wakefulness or sleep deprivation are associated with a sleep promoting function. This has been demonstrated in numerous studies in both mammals and flies. Some examples of these components include adenosine, cytokines, and NFkB.

5. In the discussion, the authors seemed to have difficulty explaining their conclusions in the context of past work, particularly in comparing their findings with those of Kuo et al (2010) and Shirasu-Hiza et al (2007) (lines 194-197). One critical distinction that should be made is that Kuo et al (2010) and several other studies in mammals report an acute sleep response to infection. That is, the host does increase sleep, but this response is short-lived and other effects on sleep occur as the infection progresses. In contrast, Shirasu-Hiza et al reported on effects that occurred over a period of days, rather than on immediate short-term effects on sleep. If over-expression of PGRP-Lca had any physiological effect in flies, the assumption is that this is a chronic manipulation, and so it may be possible that there is no acute response in this assay.

Experimental design

The use of a genetic approach to activate the immune response is clever and may be a useful way to evaluate the relationship between sleep, learning, and immune function. Unfortunately, the reported findings are not convincing in any of the measurements that were taken (immune function, memory, and sleep), and do not support the authors’ conclusions. One key technical issue is that the authors report that adult flies were fed 200 uM RU486 (line 103), which is less than half the 500 uM dose that is normally used in other studies (see Osterwalder et al 2001). If the authors wish to use a lower dose, they should verify that it is indeed effective at inducing expression of the UAS transgene. The low dose of RU486 may have contributed to the weak effects (if any) reported. What type of food medium was used during the behavioral assays? Other issues are detailed in the section below.

Validity of the findings

The statistical analyses used for measuring effects seemed misleading, inappropriate, or not well described:

1. The authors state that “whether RU486 was used” had no significant effect on memory score (line #164). However, they report a significant interaction between genotype and RU486 and conclude that “immune stimulation” decreases memory scores. Figure 1 shows that RU486 indeed decreases memory scores in GS1.32>PGRP-Lca flies, but it also increased memory scores in the +/PGRP-Lca controls. This suggests that one or both of these findings could have contributed to the significant F-statistic. A post-hoc analysis should have been performed to verify which of these cases contributed to the interaction between RU486 and genotype. Why weren’t straightforward pairwise comparisons of means performed between drug and vehicle?

2. This issue also holds true for the sleep analysis. The authors report sleep per hour in Figure 2, but apparently performed statistical analyses on daytime and nighttime sleep separately. Was the amount of sleep analyzed in 12h bins? The differences shown in Figure 2 are not at all convincing. While RU486 appears to reduce sleep in the GS1.32>PGRP-Lca flies, it also reduces sleep in the +/PGRP-Lca control group. Again, the results are reported as an effect of RU486 on genotype rather than a straightforward comparison between RU486 and vehicle groups.

Additional comments

The topic of this manuscript is interesting, but the experimental design and results fall short of supporting the conclusions. Of particular concern was the low dose of RU486 used in the experiments, failure to present the data (particularly the evidence supporting immune stimulation along with controls), and the lack of a convincing effect of PGRP-Lca on sleep or memory function. The authors should consider using an alternate approach to chronically activate the immune response in the absence of infection – alternate Gal4 drivers, UAS lines, or increasing the dose of RU486.

·

Basic reporting

The basic reporting is appropriate.

Experimental design

I may have missed it but I could not find any mention as to whether the authors tested olfactory sensitivity to determine if it was altered by their manipulation. If olfactory sensitivity was not evaluated then it remains possible that the result are due to changes in olfaction rather than to deficits in memory per se.

While memory was evaluated in both males and females, only sleep data are shown. Genetic manipulations frequently impact males and female sleep patterns differently. If the females do not respond or increase their sleep then the interpretation that the change in sleep is the cause of the memory deficit would need to be modified. These experiments can be conducted very quickly in a matter of two to three weeks.

If I understand correctly, the authors conducted a series of one-way ANOVAs. It seems to me that they should have conduced a 2(RU,Vehicle) X 3(Genotype ANOVA. Is there a particular reason for not conducting the obvious two-way ANOVA? I understand that the way the variance is portioned in this type of experiment may make it difficult to obtain a significant omnibus F for a 2-way ANOVA. If this is the case, it is worth mentioning the fact in the method section since this question frequently comes up in review.

I know that the sleep data was mostly described as being not significant. Still, I think that the basic sleep parameters should either be shown graphically or the values listed (sleep bout duration during the day and during the night as well as sleep latency). It isn't possible for the reader to determine whether these data are similar to what is seen in other genotypes or if it is vastly different. If the data are vastly different, the interpretation of the results will be altered. If they are within ranges seen in other papers then the literature will be advanced.

Validity of the findings

The data seems robust. As mentioned, sleep data in females seems to be necessary and shouldn't impose a burden to collect or present.

Additional comments

I may have missed it but I could not find any mention as to whether the authors tested olfactory sensitivity to determine if it was altered by their manipulation. If olfactory sensitivity was not evaluated then it remains possible that the result are due to changes in olfaction rather than to deficits in memory per se.

While memory was evaluated in both males and females, only sleep data are shown. Genetic manipulations frequently impact males and female sleep patterns differently. If the females do not respond or increase their sleep then the interpretation that the change in sleep is the cause of the memory deficit would need to be modified. These experiments can be conducted very quickly in a matter of two to three weeks.

If I understand correctly, the authors conducted a series of one-way ANOVAs. It seems to me that they should have conduced a 2(RU,Vehicle) X 3(Genotype ANOVA. Is there a particular reason for not conducting the obvious two-way ANOVA? I understand that the way the variance is portioned in this type of experiment may make it difficult to obtain a significant omnibus F for a 2-way ANOVA. If this is the case, it is worth mentioning the fact in the method section since this question frequently comes up in review.

I know that the sleep data was mostly described as being not significant. Still, I think that the basic sleep parameters should either be shown graphically or the values listed (sleep bout duration during the day and during the night as well as sleep latency). It isn't possible for the reader to determine whether these data are similar to what is seen in other genotypes or if it is vastly different. If the data are vastly different, the interpretation of the results will be altered. If they are within ranges seen in other papers then the literature will be advanced.

Reviewer 3 ·

Basic reporting

This manuscript has at its core a simple story – activation of the imd pathway changes sleep in alters learning. The introduction and discussion overblow the importance the work and it might be best to just describe what happened and put it in a bit of context and leave it at that.

Details:

“We establish Drosophila melanogaster as a tractable model in this field by demonstrating…” I think that Drosophila was established as a model in this field long ago. There have been a variety of papers linking sleep, feeding, climbing and cold recovery to the immune response. To say that this manuscript is the one that finally established Drosophila as a tractable model goes to far – just stick with the data.

The authors present a contrast between the work of Kuo and Shirasu-Hiza as seeing opposite effects. The problem is that the systems are so different it is hard to compare them. Kuo worked with flies in constant light, which disrupts rhythms, and found that an infection made them sleep in for a few hours the morning after the infection. Shirasu-HIza worked with cycling flies in the dark and found that sleep was disrupted for days following the infection. She didn’t even look at the morning after infection and who knows, the sleeping-in data could be buried in her data set. It doesn’t matter because they were looking at different things. The Shirasu-Hiza work doesn’t agree or disagree with the increased immune gene transcription or resistance papers because all authors were looking at different infections and they can’t be compared. Shirasu-Hiza’s work focuses on hemocytes, which don’t really show up in microarrays because they don’t make up a lot of the mass of the fly. Only Shirasu-Hiza used actual pathogens when infecting flies.

“PGRP-Lca is a pattern recognition protein that recognizes gram-negative bacteria” This is wrong – it recognizes DAP type peptidoglycan which is found on Gram negative and Gram positive bacteria. Reviews might take a short cut but it is best to stick with the actual elicitor.

The zone of inhibition assay is a decent one for measuring antimicrobial activity but unfortunately it is seldom used in Drosophila and Drosophila immunologists will inevitably ask for gene expression analysis to understand which antimicrobial peptide genes were induced. I like the inhibition assay and think that it should stand. That said, I want to see the data. Why isn’t it in the manuscript? I want to see the standard error. What does the antimicrobial activity towards some random microbe (Arthrobacter globiformis) that wasn’t used in the experiment matter?

There is a reasonably sized literature on the effects of the immune response on learning and memory in insects, in particular, honey bees and it would be good to cite this to show that immunity and memory have been seen to overlap in insects.

The authors choose to look at pure immune activation rather than an infection with an actual pathogen and argue that this is an advantage. I think this is a mistake as pathology is also an inducer of symptoms and the authors risk missing important symptoms by giving the flies an artificial and weak stimulus. I’m curious – how was the dose of the drug determined and was a dose response curve performed? Are the authors working at the top of a sigmoid dose response curve or somewhere on the slope, which would lead to more variance?

“This suggests that if type of infection were the cause of the discrepancies, our results would have just mirrored those of Kuo…” This is incorrect. Infections are complex things that signal through several pathways. Activating just one signaling pathway is never going to completely replicate the response to an actual infection. The authors are also forgetting that Shirasu-Hiza used a pathogen that largely activates the Toll signaling pathway, though it works some through imd, while Kuo used a microbe that was imd biased. Here the authors only activated imd and shouldn’t expect to perfectly replicate either experiment.

Experimental design

See Above

Validity of the findings

See above

Additional comments

See Above

---

## Round 0.2 · accepted · Accept

Thank you for responding so thoroughly to the reviewers' comments in your revision and rebuttal. The revision addresses all concerns, and I'm very pleased that we may move ahead with publication of your paper.